# Exploring Nitric Oxide as a Regulator in Salt Tolerance: Insights into Photosynthetic Efficiency in Maize

**DOI:** 10.3390/plants13101312

**Published:** 2024-05-10

**Authors:** Georgi D. Rashkov, Martin A. Stefanov, Ekaterina K. Yotsova, Preslava B. Borisova, Anelia G. Dobrikova, Emilia L. Apostolova

**Affiliations:** Institute of Biophysics and Biomedical Engineering, Bulgarian Academy of Sciences, Acad. G. Bonchev Str., Bl. 21, 1113 Sofia, Bulgaria; megajorko@abv.bg (G.D.R.); martin_12.1989@abv.bg (M.A.S.); ekaterina_yotsova@abv.bg (E.K.Y.); pborisova@bio21.bas.bg (P.B.B.); aneli@bio21.bas.bg (A.G.D.)

**Keywords:** chlorophyll fluorescence, photosystem II, oxygen-evolving complex, photosynthetic function, membrane fluidity

## Abstract

The growing issue of salinity is a significant threat to global agriculture, affecting diverse regions worldwide. Nitric oxide (NO) serves as an essential signal molecule in regulating photosynthetic performance under physiological and stress conditions. The present study reveals the protective effects of different concentrations (0–300 µM) of sodium nitroprusside (SNP, a donor of NO) on the functions of the main complexes within the photosynthetic apparatus of maize (*Zea mays* L. Kerala) under salt stress (150 mM NaCl). The data showed that SNP alleviates salt-induced oxidative stress and prevents changes in the fluidity of thylakoid membranes (Laurdan GP) and energy redistribution between the two photosystems (77K chlorophyll fluorescence ratio F_735_/F_685_). Chlorophyll fluorescence measurements demonstrated that the foliar spray with SNP under salt stress prevents the decline of photosystem II (PSII) open reaction centers (qP) and improves their efficiency (Φexc), thereby influencing Q_A_^−^ reoxidation. The data also revealed that SNP protects the rate constants for two pathways of Q_A_^−^ reoxidation (k_1_ and k_2_) from the changes caused by NaCl treatment alone. Additionally, there is a predominance of Q_A_^−^ interaction with plastoquinone in comparison to the recombination of electrons in Q_A_ Q_B_^−^ with the oxygen-evolving complex (OEC). The analysis of flash oxygen evolution showed that SNP treatment prevents a salt-induced 10% increase in PSII centers in the S_0_ state, i.e., protects the initial S_0_–S_1_ state distribution, and the modification of the Mn cluster in the OEC. Moreover, this study demonstrates that SNP-induced defense occurs on both the donor and acceptor sides of the PSII, leading to the protection of overall photosystems performance (PI_ABS_) and efficient electron transfer from the PSII donor side to the reduction of PSI end electron acceptors (PItotal). This study clearly shows that the optimal protection under salt stress occurs at approximately 50–63 nmoles NO/g FW in leaves, corresponding to foliar spray with 50–150 µM SNP.

## 1. Introduction

Numerous factors in the natural environment impact the growth and development of plants [1]. Salinity, an abiotic factor, significantly influences crop productivity. Climate changes are causing salinized regions to expand at a rate of 10% annually [2]. Salt stress disrupts plant homeostasis via two mechanisms. Initially, high soil salt concentrations hinder root water absorption [3], leading to stroma closure and reduced photosynthesis efficiency [4]. The second phase of salt stress is triggered by harmful ions that damage the structure and functions of cell membranes, causing growth inhibition and developmental changes in plants [5]. These primary effects directly lead to oxidative stress, characterized by the intense production and accumulation of reactive oxygen species (ROS, such as H_2_O_2_, O_2_^•^, OH, and ^1^O_2_), which harm proteins, lipids, and nucleic acids [3].

The chloroplasts serve as the primary sites for generating ROS, a process that relies on the interactions of light and chlorophylls. The major sites in chloroplasts that largely produce ROS are photosystem I (PSI) and photosystem II (PSII) [6]. Increased ROS causes oxidative injury to several photosynthetic enzymes and thylakoid membranes, which in turn reduces CO_2_ uptake, slows down plant growth, and ultimately lowers crop yields. The system for scavenging ROS comprises both enzymatic and non-enzymatic antioxidants [7]. The enzymatic antioxidants include superoxide dismutase (SOD), ascorbate peroxidase (APX), guaiacol peroxidase (GPX), catalase (CAT), and glutathione reductase (GR). The activity of these enzymes is essential for mitigating harmful ROS levels within cells [8]. Plants also adapt to elevated salt levels by enhancing their tissue osmotic potential by both inorganic and organic solutes [7,9,10].

Photosynthesis, the primary source of materials and energy for plant growth and development, is significantly affected by salinity [11,12]. This has sparked a growing interest in enhancing photosynthetic tolerance to boost plant yields under stressful conditions [13]. When plants are exposed to salt stress, changes occur in the thylakoid membranes, key components within the chloroplast responsible for photosynthetic light reactions. Salinity alters the number and structural arrangement of chloroplasts, leading to an increase in plastoglobules and a reduction in the thylakoids in granum in the leaf’s epidermal chloroplasts [14]. Such observations have been noted in various higher plants, including *Sulla coronaria* [15], *Thellungiella salsuginea* [16], and *Cucumis sativus* [17]. Salt stress also affects photosynthesis by altering different enzymes, leaf pigment content [18], and the structural organization of the pigment–protein complexes in the thylakoid membranes. Changes have been observed in the oxygen-evolving complex (OEC) [19,20,21], light-harvesting complex of PSII (LHCII) [22,23,24], and D1 core protein of PSII [25,26,27]. These alterations impact the electron transfer from Q_A_ to Q_B_, inhibit OEC, and affect the rate of electron transport in the thylakoid membranes [5,28,29,30].

Nitric oxide (NO) is a molecule with unpaired electrons, making it paramagnetic. It possesses the capability to readily diffuse across membranes [31]. Currently, NO is recognized as a crucial molecule that plays a role in redox signaling. It contributes to the regulation of numerous physiological processes and plays a significant role in regulating plant responses to abiotic stress [32,33]. Sodium nitroprusside (SNP) is a widely used NO donor that is used to study the role of NO in plants under physiological conditions [34] and to simulate the harmful environmental impact [35,36]. It has been shown that NO influences seed germination, maintains water balance, regulates gene expression and osmolyte accumulation, and enhances the activities of antioxidant enzymes in plants under stress. Additionally, NO directly neutralizes ROS under stressful conditions [37,38].

Over the past decade, there has been considerable interest in clarifying the role of NO in the salt tolerance of plants. NO donors can be applied to plants via spraying, incorporation into irrigation water, or injection into leaf apoplasts [32]. The protective role of NO, using an SNP donor under salinity, has been shown in different plants, such as tomato [39], cucumber [40], orange [41], cotton [42], alfalfa [43], apple [44], wheat [45], lentil [46], and sorghum [47]. Previous studies have revealed that the application of SNP under salt stress alleviates salt-induced effects on the stomatal behavior, cell water status, chlorophyll content, membrane damage, and membrane lipid peroxidation, and it also aids in the accumulation of proline, phenolic compounds, and antioxidants in plants [48,49,50].

Nitric oxide also has a direct effect on the photosynthetic electron transport, with binding sites within PSII. These include the non-heme iron located between the quinone acceptors Q_A_ and Q_B_, the Tyr YD residue, and the Mn cluster of OEC [51,52]. However, data on the influence of NO on the photosynthetic machinery are quite contradictory. It has been observed that the application of SNP (200–1000 μM) led to a decrease in the maximum quantum efficiency of PSII (Fv/Fm) and the photochemical quenching (qP) in leaves of pea and potato plants under non-stress conditions [53,54]. Previous research [55] showed that SNP (100 μM) enhanced the maximum quantum efficiency of PSII (Fv/Fm) and the effective quantum yield of PSII (Φ_PSII_) during the light-induced greening process in barley seedlings. It has also been shown that NO alleviated the salt-induced changes in the leaf area, plant dry matter production, and pigment content, as well as improved the uptake and transport of numerous macro- and micronutrients [56,57]. It has been demonstrated that NO decreases many of the negative effects of salt stress on plant photosynthetic machinery [58]. Nitric oxide decreased the salt-induced deactivation and breakdown of the PSII reaction center and enhanced their performance in salt-exposed pea, bermudagrass, soybean, and sorghum [47,58,59,60,61]. The NO-induced reduction in the harmful effects of high salt concentrations could result from the protection of the photosynthetic pigments, dissipation of surplus energy, increase in the photosystem II (PSII) quantum yield [52], and enhancement of the electron flux to the acceptor side of the photosystem I (PSI) [47]. A recent investigation conducted on *Kandelia obovata* showed that treatment with SNP under salt stress elevated endogenous NO levels, decreased ion toxicity, improved nutrient homeostasis and gas exchange parameters, and stimulated the activities of antioxidant enzymes [62].

Despite the numerous studies on the influence of NO on photosynthesis, knowledge about its defensive effects on the photosynthetic apparatus under salt stress is insufficient. The very harmful effect of salinity on the PSII complex is well known, which corresponds with the inhibition of the electron transport chain. We hypothesize that investigating the influence of SNP, as a donor of NO, at the donor and acceptor side of PSII will provide new information about the protective effect of NO under salt stress in maize plants. In the current research, we examined the impact of different concentrations of SNP on maize plants (*Zea mays* L. Kerala) cultivated in the presence of 150 mM NaCl (severe salinity). This study assessed the primary processes of photosynthesis (with PAM chlorophyll fluorescence and JIP test) and the PSII photochemical activity, revealing the impact of SNP under salt stress on the function of the photosynthetic apparatus. In addition, we measured the pigment composition, membrane stability, and levels of oxidative stress markers. The experimental results clearly show the protective mechanisms of SNP on the photosynthetic apparatus and its functions at applied SNP concentrations of 50 µM and 150 µM. These findings will contribute to offering promising information for developing strategies to increase crop resistance in saline environments.

## 2. Results

### 2.1. Pigment Content

The effects of the SNP (25–300 µM) on the pigment composition in maize leaves under salt stress (150 mM NaCl) are shown in Figure 1. The results demonstrate that the treatment with NaCl decreased the amount of chlorophylls (by 40%) and carotenoids (by 43%) (Figure 1). The simultaneous exposure of SNP and NaCl alleviated the salt-induced reduction in the pigment content, but the amounts remained lower than those of the control plants. The protection was better after application of 50 µM and 150 µM SNP in comparison to the lowest (25 µM) and the highest (300 µM) concentrations of SNP (Figure 1). The changes in the pigment composition caused a slight decrease in the Car/Chl ratio after treatment with 150 mM NaCl alone and co-treatment with NaCl and 300 μM SNP (Appendix A).

### 2.2. Stress Markers

The determination of lipid peroxidation (corresponding to MDA content) and the amount of H_2_O_2_ was used to evaluate the protective effects of SNP in maize plants under salt stress (150 mM NaCl) (Figure 2). Data showed that the amounts of MDA and H_2_O_2_ increased by about 64% and 62%, respectively, after treatment with 150 mM NaCl alone. The combined treatment with SNP and NaCl reduced the content of these stress markers (MDA and H_2_O_2_); however, these amounts were higher than the untreated plants (Figure 2a). The protective effect of SNP is less pronounced in plants treated with a concentration of 300 µM. The accumulation of H_2_O_2_ in leaves was also visualized histochemically by staining with diaminobenzidine (DAB), which forms a brown precipitate with H_2_O_2_ in a peroxidase-catalyzed reaction [63]. Data also revealed that H_2_O_2_ accumulation was in the whole leaf after 150 mM NaCl exposure, while SNP at concentrations of 25 to 150 µM significantly decreased this H_2_O_2_ accumulation (Figure 2b).

### 2.3. Membrane Stability Index and Membrane Fluidity

The membrane stability index (MSI) was used as an indicator for the impact of SNP on the membrane stability of maize leaves under salt stress. In comparison to the control, the 150 mM NaCl treatment alone reduced MSI by about 40%. The combined treatment with all studied SNP concentrations and NaCl increased the MSI compared to the NaCl treatment alone (Figure 3). The smallest protective effect was observed after co-treatment with 300 µM SNP and NaCl.

The fluidity of isolated thylakoid membranes from all variants was evaluated by general polarization (GP) of a fluorescent lipophilic membrane dye Laurdan [64]. The experimental results reveal that the salt stress leads to a decrease in the GP value, i.e., an increase in the fluidity of thylakoid membranes. The SNP application fully prevents salt-induced changes in the membrane fluidity, as the GP values of Laurdan were similar to the thylakoid membranes from the control plants (Table 1).

### 2.4. Energy Transfer between Pigment–Protein Complexes

Chlorophyll fluorescence spectra at a low temperature of 77 K were employed to assess the transfer of energy between pigment–protein complexes within the thylakoid membranes. The spectra of all studied variants were characterized with bands at 685 nm and 735 nm associated with the PSII complex and PSI complex, respectively [65]. The ratio F_735_/F_685_ reflects the redistribution of energy between both photosystems. This ratio increased by 14% after treatment with NaCl alone, but its values were similar to the control after co-treatment with all studied SNP concentrations and NaCl (Table 1).

### 2.5. PAM Chlorophyll Fluorescence

The application of 150 mM NaCl to the maize affected several parameters of PAM chlorophyll fluorescence, including the quantum yield of photochemical to non-photochemical processes (Fv/Fo), the rate of photosynthesis (R_Fd_), the excitation efficiency of open PSII centers (Φexc), the photochemical quenching (qP), and the excess excitation energy (EXC) (Figure 4). The treatment with NaCl alone led to a twofold reduction in the ratio of the quantum yield of photochemical to non-photochemical processes (Fv/Fo) and the photochemical quenching (qP). Some decrease in the rate of photosynthesis (R_Fd_, by 24%) and the excitation efficiency of open PSII centers (Φexc, by 35%) was also observed. The SNP application diminished the salt-induced decrease in the Fv/Fo, qP, Φexc, and R_Fd_. The values of parameters Φexc and R_Fd_ were similar to those of the untreated plants after co-treatment with NaCl and the concentrations of SNP up to 150 μM (Figure 4). Data also revealed that salt treatment increased the excess excitation energy (EXC) by 41% more than the control plants. The treatment with all studied SNP decreased EXC values (Appendix A).

The dark relaxation of chlorophyll fluorescence after a saturating light pulse in dark adapted leaves in both treated and untreated maize plants can be fitted by two components, with the amplitude A_1_ (fast component) and A_2_ (slow component) with rate constant k_1_ and k_2_, respectively. The constant k_1_ decreased after treatment with 150 mM NaCl alone, while k_2_ was slightly increased. After the co-treatment with SNP and NaCl, the constant k_1_ and k_2_ were similar to those of the control plants, except k_2_ in plants treated with 25 µM SNP. The data revealed also that the ratio of two components (A_1_/A_2_) increases from 22% to 33% after co-treatment with SNP concentrations up to 150 µM and NaCl compared to the untreated plants (Table 2).

### 2.6. Chlorophyll Fluorescence Induction

Selected parameters of the chlorophyll fluorescence induction used to investigate the effects of SNP under salt stress in maize were as follows: ψEo—efficiency of the electron transfer further than Q_A_^−^; N—maximum turnover of Q_A_ reduction until Fm reached (corresponding with the size of the plastoquinone pool); Vj—variable fluorescence at the J-step (corresponding with changes in the PSII acceptor side); φPo—maximum quantum yield for primary photochemistry; φRo—quantum yield for reduction of end electron acceptors at the PSI acceptor side; δRo—efficiency with which an electron from the intersystem electron carriers is transferred to reduce end electron acceptors at the PSI acceptor side; PI_ABS_—performance index for energy conservation from photons absorbed by PSII to the reduction of intersystem electron acceptors; PItotal—performance index for energy conservation from photons absorbed by PSII until the reduction of PSI end electron acceptors; RC/DIo—the reversed parameter of DIo/RC, corresponding with dissipated energy flux per RC; Wk—ratio of the J step to K step, corresponding with the changes in the PSII donor side. Data showed that NaCl treatment leads to a slight increase in the parameters Wk, Vj, and δRo, while the parameters ψEo, N, φPo, φRo, RC/DIo, PI_ABS_, and PItotal decreased. The effects were more pronounced for parameters Vj, RC/DIo, PI_ABS_, and PItotal. The foliar application of SNP under salt stress decreased the effects of NaCl on the selected JIP parameters, as the effects were better at 50 µM and 150 µM SNP (Figure 5).

### 2.7. Photochemical Activity of PSII and Flash Oxygen Evolution

The assessment of the PSII-mediated electron transport, with the electron acceptor BQ (H_2_O → BQ), was conducted to evaluate the photochemical activity of PSII. Data revealed that NaCl treatment, inhibited the PSII-mediated electron transport by 36% (Figure 6). The application of SNP alleviated the impact of NaCl on the PSII activity. This activity was the same as in the control plants after application of concentrations of 50 µM and 150 µM SNP (Figure 6).

The analysis of the flash-induced oxygen yields showed that the active PSII centers in the initial S_0_ state (S_0_ % = 100 − S_1_) and the misses (α) increased significantly after applying salt stress (Table 3). On the other hand, the SNP foliar application mitigated the salt-induced alterations in these kinetic parameters (S_0_ and α) (Table 3). The double hits (β) showed no statistically significant differences during all applied treatments.

### 2.8. NO Content

The data indicated that the application of NaCl led to an increase in the NO amount by 17% when compared to the untreated plants (Figure 7). The co-treatment with SNP under salt stress caused an additional rise in NO content depending on the applied SNP concentration (Figure 7).

### 2.9. Principal Component Analysis

Principal component analysis (PCA) revealed that the first two components explain 99.43% of the data variability (Appendix A). The control maize, positioned in the first quadrant, shows a negative correlation with the EXC parameter describing the dissipation of excess energy, which is located in the third quadrant. A strong positive correlation was determined for the parameters EXC and F_735_/_685_ and the maize treated with 150 mM NaCl, located in the lowest point of the second quadrant. A more pronounced positive correlation of the photochemical to non-photochemical processes (Fv/Fo) (first quadrant, far from PC1 and PC2 axes) and a weaker one for the photosynthetic rate (R_Fd_) (second quadrant near the PC1 axis) was found with respect to the control maize in comparison to the other plant variants (150 mM NaCl and 150 μM SNP + 150 mM NaCl), located in the second quadrant. The variables located in the fourth quadrant are related to pigment content (Car, Chl, Car/Chl), membrane fluidity (Laurdan GP), and photochemical activity (qP, Φexc), and have an insignificant contribution to the changes occurring in all the maize variants, with a weakly pronounced positive correlation with the control maize plants and a negative correlation with the maize treated with 150 mM NaCl alone.

## 3. Discussion

Despite extensive research, the precise role of NO in plant survival under adverse environmental conditions has not been fully understood, particularly regarding its defensive function related to the operation of various pigment–protein complexes in photosynthetic membranes. In this study, we reveal new evidence regarding the impact of SNP, acting as a NO donor, on different components of the photosynthetic apparatus under salt stress.

Data in this study demonstrated a reduction in photosynthetic pigments (Chl and Car) similar to previous studies on wheat, sorghum, pea, barley, and other plant species [66,67,68,69,70,71]. This negative impact on the pigments is due to an inhibition of the pigment biosynthesis and/or an enhancement of their degradation [72]. The protective effect of 50 µM and 150 µM SNP on the photosynthetic pigments (Figure 1) may result from stimulating Chl and Car biosynthesis observed previously [34]. Additionally, SNP could counteract the salt-induced negative effect on their biosynthesis, which is accompanied by enhancement of the LHCII accumulation [55,73]. The data also demonstrated that foliar spray with SNP prevents, under salt stress, the decrease in the Car/Chl ratio at concentrations up to 150 µM SNP (Appendix A). Considering, the important role of carotenoids as antioxidants involved in the protection of thylakoid membranes from oxidative stress at a high salinity [74], it could be suggested that their enhancement is also one of the protective mechanisms of SNP under salt stress.

The enhancement of the H_2_O_2_ content (Figure 2) and other ROS species caused lipid peroxidation and disruption of the membrane structures [75,76]. The high accumulation of the MDA under salt stress revealed an enhancement of the lipid peroxidation (Figure 2). In addition to the lipid peroxidation, the salt stress also leads to changes in the lipid composition [77]. The salt-induced changes in the lipids and the proteins cause structural changes in the thylakoid membranes, corresponding with the reduction in thylakoids in grana regions [14,16,78], which is accompanied by a decrease in the MSI (Figure 3).

Furthermore, the observed decrease in Laurdan GP values for the thylakoid membranes isolated from NaCl-treated plants compared to controls (Table 1) clearly indicates an increase in membrane fluidity. It has been suggested that the more fluid thylakoid membranes may facilitate the diffusion of the LHCII complex from PSII to PSI complexes [79,80], which is confirmed by an increase in the fluorescence emission ratio (F_735_/F_685_) after NaCl treatment (Table 1). Our results reveal that the application of the SNP not only decreased the salt-induced oxidative stress (i.e., changes in the amounts of H_2_O_2_ and MDA) but also prevented the membrane structural alterations, such as changes in the membrane fluidity and energy transfer between both photosystems (Figure 3 and Table 1). In addition, current data demonstrated that SNP foliar spray (at concentrations of 25 to 150 µM) alleviated salt-induced oxidative stress to a certain extent. This reduction was evident through the decreased amounts of H_2_O_2_ and MDA in salt-stressed leaves, thus preventing salt-induced damage of the membranes (Figure 3).

Previous studies [7,12,17,30,81,82,83,84], as well as the data in the present study, revealed a strong influence of salinity on the function of the photosynthetic apparatus. The analysis of the chlorophyll fluorescence curves at room temperature demonstrated that the salt treatment led to a decrease in the ratio of the photochemical to non-photochemical processes (Fv/Fo), the photochemical quenching (qP), the excitation efficiency of open PSII centers (Φexc), and the rate of photosynthesis (R_Fd_) (Figure 4). This impact on the PSII function is due to changes in the acceptor and donor side of the complex [30,85,86]. At the same time, the parameter EXC showed an increase in the energy losses (Appendix A), which was accompanied with a decrease in the efficiency of PSII under NaCl treatment alone (Figure 4). The salt treatment also led to a decrease in the parameter RC/DIo (the reversed parameter of the dissipated energy flux per RC, DIo/RC), showing an increase in the dissipated energy (Figure 5). Dissipation of the excess light corresponds with a decrease in ROS formation, acting as a photoprotective mechanism [87].

To assess the salt-induced changes and the protective mechanisms of SNP on the PSII acceptor side, we analyzed the chlorophyll fluorescence signals following a saturating light pulse [88,89]. Data revealed that the alterations in the PSII functions were a result of salt-induced impacts on both pathways of Q_A_^−^ reoxidation: one involving plastoquinone and the other via recombination of electrons in Q_A_ Q_B_^−^ with oxidized S_2_ (or S_3_) states of the OEC (Table 2). However, the influence on the Q_A_ functions decreased the efficiency of the electron movement further than Q_A_^−^ (ψEo parameter) and the probability that an electron from the intersystem electron carriers is transferred to reduce end electron acceptors at the PSI acceptor side (Figure 5). The performance index on the absorption base (PI_ABS_), commonly used to assess overall PSII performance [83,90], was strongly influenced by NaCl (Figure 5), corresponding to the inhibition of PSII-mediated electron transport (Figure 6). Moreover, PItotal was also reduced under salt stress, which suggests delayed performance from the PSII electron donor side to the reduction of the PSI end electron acceptors.

In SNP-treated plants under salt stress, Q_A_ interaction with plastoquinone prevailed, resulting in an increased A_1_/A_2_ ratio (Table 2). These changes were associated with an increase in open PSII centers (qP) and their efficiency (Φexc), improving the photosynthetic performance (PI_ABS_) and performance of the electron transport reduction of the PSI end electron acceptors along with stimulation of the photosynthesis rate (R_Fd_) (Figure 4 and Figure 5). Simultaneously, there was an elevation in excess excitation energy (EXC) (Appendix A). All these observations corresponded with the inhibition of PSII-mediated electron transport in the presence of the exogenous acceptor BQ, after NaCl treatment alone and with full protection, at applied concentrations of 50 µM and 150 µM SNP (Figure 6).

More detailed information regarding the influence of NaCl on the PSII donor side was obtained by analyzing kinetic parameters of the flash-induced oxygen evolution without exogenous acceptors, i.e., electrons are accepted from the plastoquinone (PQ) (Table 3). The salt treatment resulted in changes in the initial S_0_–S_1_ state distribution due to an increase in the number of active PSII centers in the most reduced S_0_ states (Mn^2+^, Mn^3+^, Mn^4+^, Mn^4+^). This increase indicates a modification in the Mn_4_Ca cluster within the OEC [91,92]. A similar influence of salt stress on PSII centers in the initial S_0_ state in darkness has also been shown in barley plants [30,93]. The observed increase in the S_0_ state corresponds with an increase in the misses (α). The application of SNP under salt stress fully prevented the salt-induced alterations in the initial S_0_–S_1_ state distribution of the PSII complexes. Therefore, the SNP protection on the PSII donor side is most probably due to the prevention of the salt-induced modification of the Mn_4_Ca cluster of OEC.

These effects of NO on the membrane integrity and function of the photosynthetic apparatus could also result from direct neutralization of the ROS, enhancement of antioxidant enzyme activities, and increased accumulation of osmolyte compounds [38]. The application of SNP protected against membrane damage, as well as the salt-induced changes in membrane fluidity, Q_A_ reoxidation, and modification of the Mn clusters of the OEC. By preventing the salt-induced changes in both donor and acceptor sides of PSII, the PSII performance and overall function of the photosynthetic apparatus were improved. The data also revealed that better protection of the thylakoid membranes is achieved at a concentration of NO up to 63 nmoles/g FW (at 150 µM SNP).

## 4. Materials and Methods

### 4.1. Plant Growth Conditions and Treatment

The seeds of maize (*Zea mays* L. Kerala) were obtained from Euralis Ltd. in Lescar, France. The plants were cultivated hydroponically in a climate chamber under controlled conditions: 25 °C (daily)/22 °C (night) temperature, 150 μmol photons/m^2^ s light intensity, a 12 h light/dark photoperiod, and 70% air humidity. Two-week-old maize plants were foliar-sprayed with different concentrations of SNP (25 μM, 50 μM, 150 μM, and 300 μM) 24 h before the addition of 150 mM NaCl in the Hoagland solution. The solution of SNP can be released as free nitric oxide (NO) or NO^+^ and free CN^−^ or CN radicals [53,94]. The measurements were performed 5 days after addition of NaCl in nutrient solution. The concentrations of NaCl and SNP are based on a preliminary study [34,47]. The following variants were studied: control (without SNP and NaCl), NaCl (150 mM NaCl), 25 SNP + NaCl (25 μM SNP and 150 mM NaCl), 50 SNP + NaCl (50 μM SNP and 150 mM NaCl), 150 SNP + NaCl (150 μM SNP and 150 mM NaCl), 300 SNP + NaCl (300 μM SNP and 150 mM NaCl). Two independent experiments were conducted in four replications for each variant (four boxes with three plants in each). The measurements were made on the fully expanded leaves.

### 4.2. Leaf Pigment Content

The method of Lichtenthaler (1987) was used to determine the amount of chlorophylls and carotenoids. Leaf tissue (0.03 g) was cut into small pieces and grinded with 8 mL of 80% acetone in dark and cold. After centrifugation at 4500× *g* for 10 min at 0–4 °C, the supernatant was measured spectrophotometrically (Specord 210 Plus, Ed. 2010, Analytik Jena AG, Jena, Germany) at 663.2, 646.8, and 470 nm and the pigment content totals Chl and Car were calculated using Lichtenthaler’s equations [95].

### 4.3. Stress Markers, Membrane Stability Index

The fully expanded leaves were taken from different variants to estimate the content of hydrogen peroxide (H_2_O_2_) and malondialdehyde (MDA) following the procedure described in [34]. The H_2_O_2_ and MDA amounts were calculated by recording the absorbance at 390 nm and 532 nm, respectively, using Specord 210 Plus (Edition 2010; Analytik Jena AG, Jena, Germany), and the values are expressed as nmol per g DW.

The visualization of the H_2_O_2_ accumulation in maize leaves was performed with the dye diaminobenzidine (DAB) because peroxidase catalyzed the reaction of DAB with H_2_O_2_ to form a brown polymer as previously described in [63]. Several fresh leaves were soaked in DAB solution (1 mg/mL) and incubated at room temperature overnight in the dark. The leaves were then placed in boiling ethanol (95%) to remove the background.

The membrane stability index (MSI) for maize leaves was evaluated based on the electrolyte conductivity (EC) as described previously in [34].The MSI values were calculated as MSI (%) = [1 − (EC1/EC2)] × 100, where EC1 and EC2 are the measured electrolyte conductivities of the leaf sample solutions after incubation for 24 h at 20 °C and after boiling for 30 min, respectively.

### 4.4. Thylakoid Membrane Fluidity

The fluidity of thylakoid membranes was followed by a fluorescence polarization study with a fluorescent lipophilic dye Laurdan (6-Dodecanoyl-2-dimethylaminonaphthalene, Sigma-Aldrich, St. Louis, MO, USA) as described previously in [64,96]. The isolated thylakoid membranes with a concentration of 15 μg Chl/mL were incubated with 30 μM Laurdan, using 1 mM stock solution dissolved in dimethyl sulfoxide, (DMSO, Sigma-Aldrich, St. Louis, MO, USA) for 30–40 min at room temperature in the dark. The steady-state fluorescence polarization was determined using a spectrofluorometer JASCO FP8300 (Jasco, Tokyo, Japan). Fluorescence was excited at 390 nm and registered at 460 and 515 nm with a 10 nm emission slit width using a quartz cuvette of 1 cm path length according to [64]. The general polarization (GP) of Laurdan fluorescence was determined as GP = (I_460_ − I_515_)/(I_460_ + I_515_), where I_460_ is the fluorescence intensity at 460 nm (characteristic for tightly packed membrane lipids) and I_515_ is the fluorescence intensity at 515 nm (characteristic for less tightly packed lipids). The lower GP values point to an increased fluidity, i.e., membranes with a less ordered fluid phase [64].

### 4.5. Low-Temperature (77 K) Fluorescence Measurements

Low-temperature (77K) chlorophyll fluorescence emission spectra were obtained using a spectrofluorometer (Jobin Yvon JY3, Division d’Instruments S.A., Longjumeau, France) equipped with a nitrogen device. The samples were quickly frozen in liquid nitrogen. The measurements were made as in [30]. Data analysis and graphing software (Origin version 9.0, OriginLab Corporation, Northampton, MA, USA) was employed to analyze emission spectra registered after the excitation of Chl *a* (at 436 nm). The spectra of all studied variants were characterized with two fluorescence bands at 685 nm (for PSII) and 735 nm (for PSI). The fluorescence emission ratio F_735/685_ was calculated. This ratio characterized the energy redistribution between both photosystems [30,65].

### 4.6. Room-Temperature Chlorophyll Fluorescence

Pulse-modulated amplitude (PAM) chlorophyll fluorescence was measured on dark-adapted (for 15 min) leaves using a fluorimeter (H.Walz, Effeltrich, Germany, model PAM 101–103). The measurements were made as in [97]. The minimal fluorescence level (Fo) was recorded at a frequency of 1.6 kHz and a measuring light of 0.110 µmol photons/m^2^s PFD. The maximal fluorescence levels for the dark-adapted state (Fm) and light-adapted state (Fm’) were determined using saturated pulse light of 3000 photons µmol/m^2^s for 0.8 s. The photosynthetic process was triggered by exposing the plants to actinic light with an intensity of 150 μmol photons/m^2^s. The PAM chlorophyll fluorescence parameters calculated to assess the impact of SNP under salt stress on the function of the photosynthetic apparatus functions are as follows [98]: Fv/Fo—the ratio of photochemical to non-photochemical processes; qP—the coefficient of photochemical quenching [98]; Φexc—the excitation efficiency of open PSII centers [99]; and EXC—the excitation excess energy [100]. The parameter R_Fd_—the chlorophyll fluorescence decay ratio after saturating light pulse (3000 μmol photons/m^2^s) in dark-adapted leaves—was determined as described in [84].

The additional information about the effects of SNP under salt stress on the PSII complex gives the dark relaxation of chlorophyll fluorescence by a signal after saturating light pulse in dark-adapted leaves. Analysis of the curves provides details about the electron transfer from Q_A_ to plastoquinone [29]. Fluorescence signals can be fitted by two components. The ratio of the fast (A_1_) and slow (A_2_) components (A_1_/A_2_) and rate constants k_1_ and k_2_ were determined as in [101]. The constants k_1_ and k_2_ are related to Q_A_—reoxidation pathways [101].

Chlorophyll fluorescence induction curves were obtained and measured with a Handy PEA+ device (Hansatech, Norfolk, UK), as described in [47]. The leaves were adapted in dark for 30 min. The light pulse intensity was 3000 μmol photons/m^2^s. The following JIP parameters were determined [29]: PI_ABS_ and PItotal—the performance indexes, Vj—relative variable fluorescence at the J step, N—maximum turnovers of Q_A_ reduction until Fm was reached, ϕ_Po_—the maximum quantum yield of primary photochemistry, ψ_Eo_—moves an electron into the electron transport chain beyond Q_A_^−^, RC/DIo—the reversed parameter of DIo/RC—dissipated energy flux per RC (at t = 0), Wk—the ratio of K phase to J phase, ϕ_Ro_—quantum yield of reduction of end electron acceptors at the PSI acceptor side, δ_Ro_—efficiency with which/probability that an electron from the intersystem electron carriers moves to reduce end electron acceptors at the PSI acceptor side. Fully developed leaves (the middle area of the third and fourth leaves) were used for all fluorescence analyses.

### 4.7. Isolation of Thylakoid Membranes

Thylakoid membranes were isolated from leaves of maize plants as described in [102]. For measurements, the thylakoid membranes were suspended in 20 mM HEPES (pH 7.6), 0.4 M sucrose, 5 mM MgCl_2_, 10 mM NaCl. The Chl content in thylakoid membranes was extracted with 80% (*v*/*v*) acetone and was assessed using Lichtenthaler’s equations [95].

### 4.8. Photochemical Activity of PSII

The photochemical activity of PSII was assessed for the PSII-mediated electron transport as in [30]. The measurements were made on the isolated thylakoid membranes (25 µg Chl/mL) using a Clark-type electrode (Model DW1, Hansatech, Instruments Ltd., Norfolk, UK). The reaction medium for PSII-mediated electron transport (H_2_O → BQ) contained 20 mM MES (pH 6.5), 0.4 M sucrose, 5 mM MgCl_2_, 10 mM NaCl, 0.4 mM BQ (1,4-benzoquinone).

### 4.9. Oxygen Evolution Measurements

The flash-induced oxygen yields under short flash illumination by saturating (4 J) and short (10 s) periodic flash sequences applied on the thylakoid membranes’ suspension (150 µg Chl/mL) were determined using a polarographic oxygen rate electrode (Joliot-type) without the addition of artificial electron acceptors as described previously [91]. The analysis of the flash-induced oxygen yields was performed using the model of Kok [103]. According to this model, the cooperation of five oxidation states of OEC (S_0_–S_4_) in the same PSII centers is required for the production of one oxygen molecule. In the darkness, only the S_0_ and S_1_ states are stable. The calculations of the used parameters were made as described in [91]: S_0_—percentage of active oxygen-evolving PSII centers in the most reduced (S_0_) state in darkness, i.e., the initial S_0_–S_1_ state distribution (S_0_ + S_1_ = 100%), misses (α), and double hits (β).

### 4.10. NO Content

The NO content was determined following the method of [104]. The leaves were homogenized in an acetic acid buffer with low pH supplemented with zinc acetate. After centrifugation, the supernatant was neutralized with charcoal, followed by the addition of Griess reagent. The NO content (nmol/g FW) was determined using a calibration curve generated with sodium nitrite as a standard.

### 4.11. Principal Component Analysis

Principal component analysis (PCA), a multivariate statistical method, was employed to reduce a vast array of measured parameters into the most informative ones [105]. PCA was utilized to explore the impact of SNP under salt stress (150 mM NaCl) on the fluorescence parameters and their correlations with bio-chemical parameters. For categorizing/classifying the variations in response to salt stress and SNP application, a clustering algorithm was implemented [106]. PCA multivariate statistical analysis was conducted, and graphical representations of PCA were generated using Originlab 9 software for data analysis and graphing (OriginLab Corporation, Northampton, MA, USA).

### 4.12. Statistical Analysis

Differences among the various t were assessed by Student’s *t*-test or one-way ANOVA with post hoc Tukey’s test. The mean values (n = 8) were considered statistically significant at least for *p* < 0.05.

## 5. Conclusions

In summary, the data in the present study revealed the protection of thylakoid membranes in maize after foliar spray with SNP under salt stress. The protection is due to a decrease in the stress markers (H_2_O_2_ and MDA) preventing the salt-induced changes in membrane fluidity and energy transfer between the pigment–protein complexes within the photosynthetic apparatus. The effects of SNP were accompanied by an increase in the PSII open reaction center and tier efficiency, as well as the prevention of salt-induced alterations on both the donor and acceptor sides of PSII. The data demonstrated that co-treatment with SNP and NaCl leads to a decrease in salt-induced changes in the rate constants of two pathways of Q_A_ reoxidation: one involving plastoquinone and the other involving recombination on Q_A_Q_B_- with oxidized S_2_ (or S_3_) states of the OEC. Moreover, the application of SNP under salt stress predominantly favors reoxidation through interaction with plastoquinone. The application of the SNP also prevented the modification of the Mn clusters of the OEC at high salt concentrations and improved the oxygen-evolving activity. The data also revealed that SNP provided better protection under salt stress at concentrations between 50 μM and 150 μM or an amount of NO equivalent to 50–63 nmoles/g FW in leaves.

## Figures and Tables

**Figure 1 plants-13-01312-f001:**
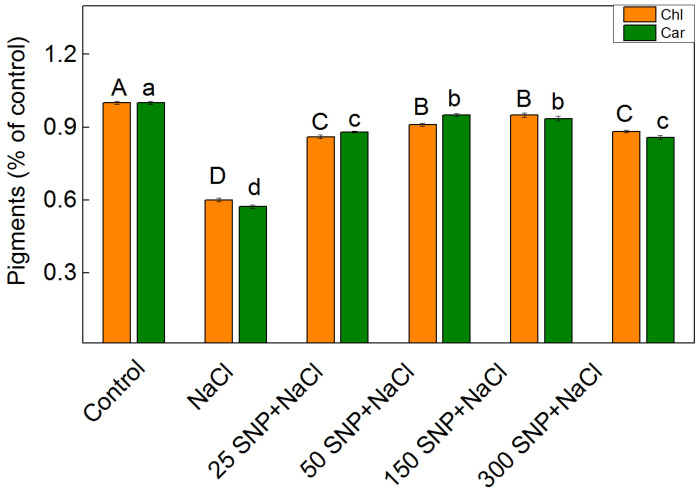
Impact of SNP on the amounts of chlorophylls (Chl) and carotenoids (Car) in maize (*Zea mays* L. Kerala) under salt stress. The control value for the chlorophylls is 49.498 mg/g DW and, for the carotenoids, it is 8.159 mg/g DW. The mean values (±SE) were determined from 8 measurements. Significant differences between variants at *p* < 0.05 are indicated by different letters (uppercase for chlorophyll and lowercase for carotenoid levels).

**Figure 2 plants-13-01312-f002:**
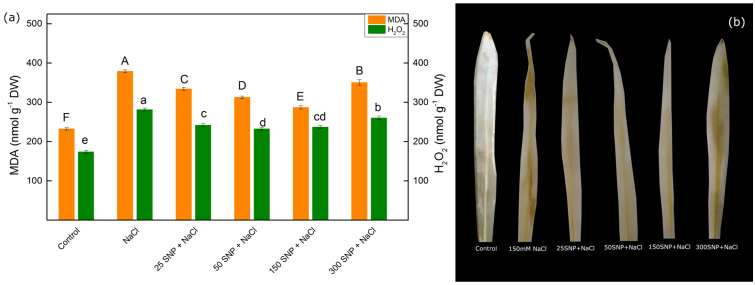
Effects of the SNP c on MDA and H_2_O_2_ contents under salt stress (150 mM NaCl) (**a**), and visualization of the H_2_O_2_ accumulation in maize leaves by DAB staining (**b**). Mean values (±SE) were determined from 8 measurements. Different letters indicate significant differences among variants at *p* < 0.05 (uppercase for MDA and lowercase for H_2_O_2_).

**Figure 3 plants-13-01312-f003:**
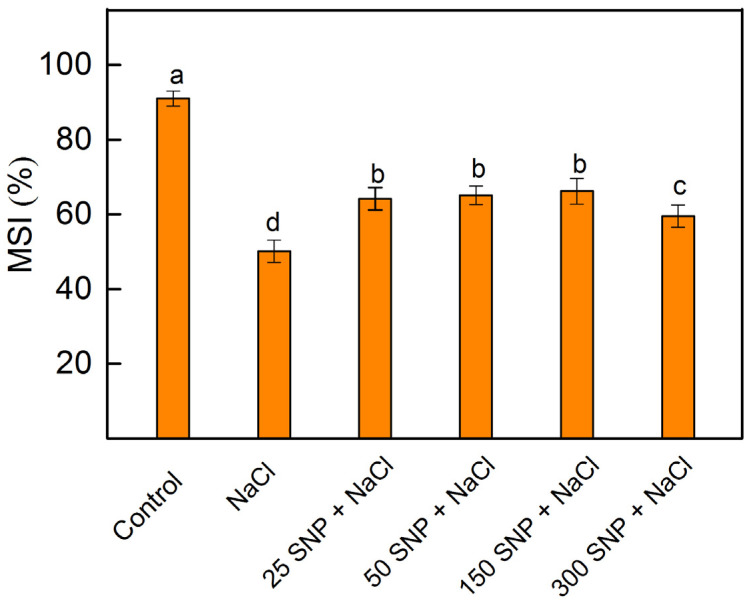
Effects of t SNP on the membrane stability index (MSI) of maize leaves (*Zea mays* L. Kerala). Mean values (±SE) were determined from 8 measurements. Different letters indicate significant differences among variants at *p* < 0.05.

**Figure 4 plants-13-01312-f004:**
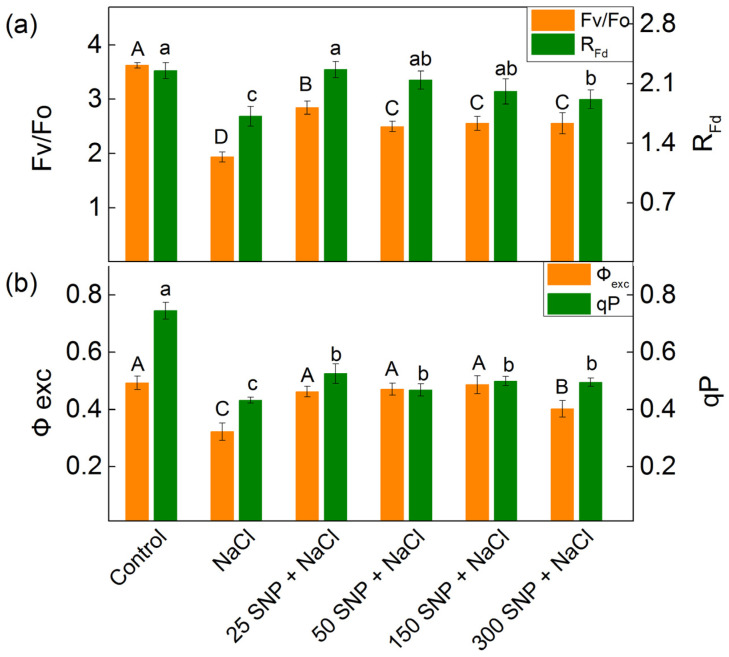
Impact of various SNP concentrations under salinity conditions (150 mM NaCl) on the PAM chlorophyll fluorescence parameters in maize (*Zea mays* L. Kerala). (**a**) Quantum yields of photochemical to non-photochemical processes (Fv/Fo), and the chlorophyll fluorescence decay ratio (R_Fd_); (**b**) the excitation efficiency of open PSII centers (Φexc) and photochemical quenching (qP). Mean values ± SE were calculated from 8 independent measurements. The different letters indicate significant differences among variants at *p* < 0.05 (uppercase for Fv/Fo and Φexc, lowercase for R_Fd_ and qP).

**Figure 5 plants-13-01312-f005:**
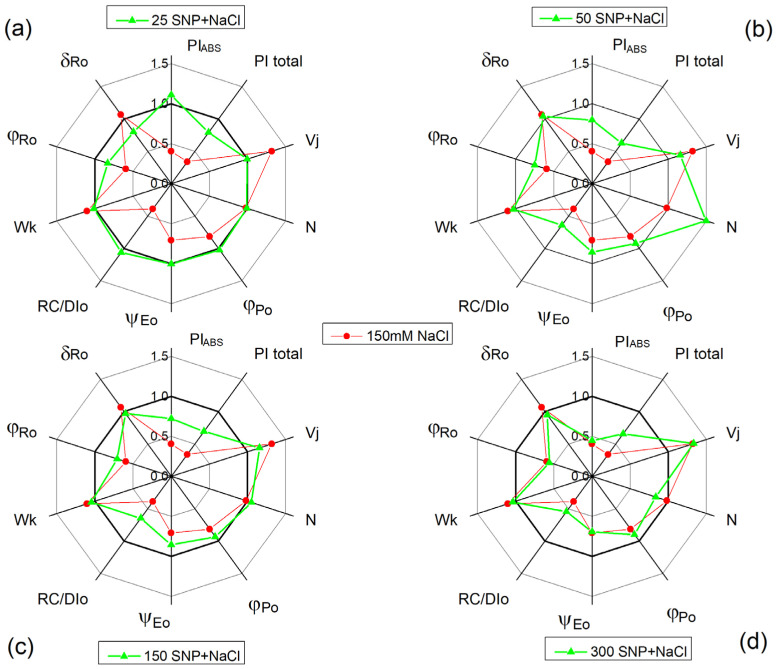
The influence of SNP in the presence of the 150 mM NaCl on the selected JIP parameters in maize (*Zea mays* L. Kerala). Red lines show the influence of 150 mM NaCl. Green lines show the influence of different concentrations of SNP under salt stress: (**a**) 25 µM SNP and 150 mM NaCl; (**b**) 50 µM SNP and 150 mM NaCl; (**c**) 150 µM SNP and 150 mM NaCl; (**d**) 300 µM SNP and 150 mM NaCl. ψEo—efficiency/probability that an electron moves further than Q_A_^−^; N—maximum turnover of Q_A_ reduction until Fm reached; Vj—variable fluorescence at the J-step (2 ms); φPo—maximum quantum yield for primary photochemistry; φRo—quantum yield for reduction of end electron acceptors at the PSI acceptor side; δRo—efficiency with which an electron from the intersystem electron carriers is transferred to reduce end electron acceptors at the PSI acceptor side (RE); PI_ABS_—performance index for energy conservation from photons absorbed by PSII until the reduction of intersystem electron acceptors; PItotal—performance index for energy conservation from photons absorbed by PSII until the reduction of PSI end electron acceptors; RC/DIo—the reversed parameter of DIo/RC—dissipated energy flux per RC (at t = 0); Wk—ratio of the J step to K step. All parameters are normalized to the parameters of the untreated plants.

**Figure 6 plants-13-01312-f006:**
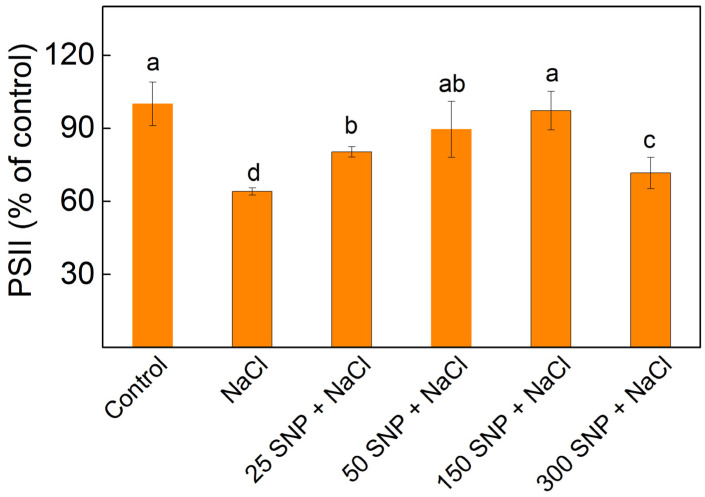
Photochemical activity of PSII (H_2_O → BQ) in isolated thylakoid membranes from leaves of maize (*Zea mays* L. Kerala) after exposure to various SNP concentrations and 150 mM NaCl. The values are expressed as a percentage of the respective control. Significant differences between variants at *p* < 0.05 are indicated by different letters.

**Figure 7 plants-13-01312-f007:**
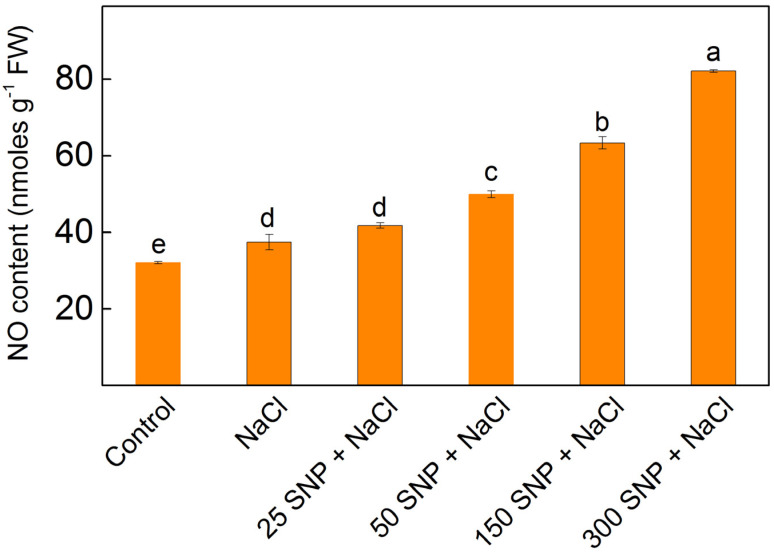
NO levels in leaves of maize (*Zea mays* L. Kerala) after treatment with different SNP concentrations in the presence of 150 mM NaCl. Significant differences between treatments at *p* < 0.05 are denoted by different letters.

**Table 1 plants-13-01312-t001:** Impact of various SNP concentrations on Laurdan GP values and the low-temperature (77 K) fluorescence emission ratio F_735_/_685_ of isolated thylakoid membranes from leaves of maize (*Zea mays* L. Kerala) grown under salt conditions (150 mM NaCl). The chlorophyll fluorescence was excited at 436 nm. Statistically significant differences at *p* < 0.05 are marked by different letters among the mean values (±SE) in the corresponding column (n = 8).

Variants	GP	F_735/685_
Control	0.478 ± 0.021 ^a^	1.25 ± 0.06 ^b^
NaCl	0.388 ± 0.018 ^b^	1.43 ± 0.06 ^a^
25 SNP + NaCl	0.451 ± 0.014 ^a^	1.28 ± 0.05 ^b^
50 SNP + NaCl	0.485 ± 0.017 ^a^	1.28 ± 0.10 ^b^
150 SNP + NaCl	0.465 ± 0.016 ^a^	1.29 ± 0.06 ^b^
300 SNP + NaCl	0.446 ± 0.014 ^a^	1.33 ± 0.03 ^b^

**Table 2 plants-13-01312-t002:** Influence of various SNP concentrations under salt stress (150 mM NaCl) on the rate constants (k_1_ and k_2_) and the ratio of the amplitudes of the fast and slow components (A_1_/A_2_) of the relaxation of chlorophyll fluorescence after saturating light pulse in dark adapted leaves of maize (*Zea mays* L. Kerala). The different letters among the mean values (±SE) in the corresponding column (n = 8) show the statistical differences at *p* < 0.05.

Variants	k_1_ (s^−1^)	k_2_ (s^−1^)	A_1_/A_2_
Control	1.741 ± 0.045 ^a^	0.085 ± 0.007 ^b^	6.731 ± 0.230 ^b^
150 mM NaCl	1.578 ± 0.049 ^b^	0.107 ± 0.010 ^a^	6.263 ± 0.221 ^c^
25 μM SNP + NaCl	1.826 ± 0.099 ^a^	0.104 ± 0.005 ^a^	8.929 ± 0.341 ^a^
50 μM SNP + NaCl	1.895 ± 0.140 ^a^	0.090 ± 0.008 ^b^	8.184 ± 0.321^a^
150 μM SNP + NaCl	1.903 ± 0.079 ^a^	0.094 ± 0.008 ^b^	8.392 ± 0.503 ^a^
300 μM SNP + NaCl	1.739 ± 0.075 ^a^	0.085 ± 0.006 ^b^	6.974 ± 0.409 ^b^

**Table 3 plants-13-01312-t003:** Effects of 150 mM NaCl treatment and different SNP concentrations on the kinetic parameters of the flash-induced oxygen yields: S_0_—the PSII centers in the initial reduced state (S_0_ % = 100 − S_1_) in the darkness, misses (α), and double hits (β). Significant differences between variants at *p* < 0.05 are indicated by different letters.

Variants	S_0_ (%)	α (%)	β (%)
Control	24.5 ± 1.3 ^b^	20.3 ± 1.8 ^c^	6.2 ± 1.3 ^a^
150 mM NaCl	34.1 ± 2.7 ^a^	28.8 ± 1.5 ^a^	7.5 ± 1.7 ^a^
25 SNP + NaCl	25.3 ± 1.2 ^b^	24.3 ± 2.1 ^b^	6.8 ± 1.4 ^a^
50 SNP + NaCl	23.3 ± 1.7 ^b^	25.7 ± 1.2 ^b^	5.9 ± 1.8 ^a^
150 SNP + NaCl	23,1 ± 1.9 ^b^	24.3 ± 2.5 ^b^	6.4 ± 1.7 ^a^
300 SNP + NaCl	24.1 ± 1.4 ^b^	25.3 ± 1.3 ^b^	6.9 ± 1.4 ^a^

## Data Availability

Data are contained within the article and Appendix A.

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
