# Peer review of "Exploring Nitric Oxide as a Regulator in Salt Tolerance: Insights into Photosynthetic Efficiency in Maize"

_plants, 2024, doi:10.3390/plants13101312_

Round 1

Reviewer 1 Report

Comments and Suggestions for Authors

The present manuscript tested the protective effects of different concentrations (0-300 µM) of sodium nitroprusside (SNP, a donor of NO) on the functions of the main complexes within the photosynthetic apparatus of maize. The present research is an interesting idea, and role of NO is increasing is under diverse stress conditions. Despite this there are some problems and shortcomings that should be improved, removed or clarified. Below I identified the diverse issues that must be addressed before any decision.

- The abstract section is written well, however, authors must add a strong rational at the start of abstract section.

- abstract section is written in very descriptive form and authors can make it attractive by adding some numerical values.

- Keywords: These keywords “: PAM chlorophyll fluorescence; JIP test” can be deleted during revisions.

- What is extent of salinity stress? The information about it can enrich the introduction section.

- A few lines could be added to explain the significance of antioxidants in tolerance to salt stress – the role of salinity stress in generating ROS?

- A wide range of citations in introduction such as the foundational work on antioxidants and photo-oxidative stress would help the manuscript in terms of the rationale behind the work.

- Lines 106-114 must be revised and here rather than long statements, authors must add the study hypothesis along with clear objectives.

- In this research, Zea mays L. Kerala was investigated. What is the need of this sentence here?

- Line 473: The treatment with NaCl was for 5 days????

- What was the experiment design used by the authors?

- The results section is a very difficult read.  Shorter simpler sentences should be used.  At the moment the text is too complex and multi-clausal making it difficult to follow. 

- I didn’t see any results that looked like a multiple comparison ANOVA.  The Materials and Methods report a two-way ANOVA to examine interactions.  The letters in the tables and above the columns of the histogram are from a one-way ANOVA with post-hoc Tukey test?  I think this needs clarifying to make it more evident to the reader.

- The discussion section contains many un-necessary results and there is need to avoid adding too many results in discussion section.

Comments on the Quality of English Language

Please recheck the paper for typos.

Author Response

Report to the comments of reviewer 1 on the manuscript (plants-2974020) titled  “ Exploring Nitric Oxide as a Regulator in Salt Tolerance: Insights into Photosynthetic Efficiency in Maize” by Rashkov et al.

The authors would like to thank the reviewer for constructive and insightful comments about this work. We considered all comments and suggestions to be justified, and corrected the manuscript accordingly. Please, find the detailed list of all edits below. The newly edited text parts are indicated with red letters.

Comments and Suggestions for Authors

The present manuscript tested the protective effects of different concentrations (0-300 µM) of sodium nitroprusside (SNP, a donor of NO) on the functions of the main complexes within the photosynthetic apparatus of maize. The present research is an interesting idea, and role of NO is increasing is under diverse stress conditions. Despite this there are some problems and shortcomings that should be improved, removed or clarified. Below I identified the diverse issues that must be addressed before any decision.

- The abstract section is written well, however, authors must add a strong rational at the start of abstract section.

- abstract section is written in very descriptive form and authors can make it attractive by adding some numerical values.

Answer: Some corrections and additions have been made in the abstract of the revised manuscript.

- Keywords: These keywords “: PAM chlorophyll fluorescence; JIP test” can be deleted during revisions.

Answer: PAM chlorophyll fluorescence; JIP test are deleted in the revised version.

- What is extent of salinity stress? The information about it can enrich the introduction section.

Answer: An addition was made for the degree of salinity in the introduction of the revised manuscript.

- A few lines could be added to explain the significance of antioxidants in tolerance to salt stress – the role of salinity stress in generating ROS?

- A wide range of citations in introduction such as the foundational work on antioxidants and photo-oxidative stress would help the manuscript in terms of the rationale behind the work.

Answer: The addition for the ROS generation under salt stress and the significance of the antioxidants in the salt tolerance of plants were made in the revised manuscript.

- Lines 106-114 must be revised and here rather than long statements, authors must add the study hypothesis along with clear objectives.

Answer: This paragraph has been changed in the revised manuscript.

- In this research, Zea mays L. Kerala was investigated. What is the need of this sentence here?

Answer: The sentence is deleted in the revised manuscript.

- Line 473: The treatment with NaCl was for 5 days????

Answer: The correction was made in the revised manuscript. We hope that the change made in "Materials and Methods" gives a clearer the treatment time.

- What was the experiment design used by the authors?

Answer: The experimental conditions in this study are based on our previous studies on the effects of different concentrations of NaCl [ Stefanov et al., Plants 2021. 10, 1469] on maize as well as the effects of the different SNP concentrations under physiological conditions [Rashkov et al., Plants 2024. 13, 118].

- The results section is a very difficult read.  Shorter simpler sentences should be used.  At the moment the text is too complex and multi-clausal making it difficult to follow. 

Answer: Thank you for the remark. Based on your suggestion, corrections have been made to the results section.

- I didn’t see any results that looked like a multiple comparison ANOVA.  The Materials and Methods report a two-way ANOVA to examine interactions.  The letters in the tables and above the columns of the histogram are from a one-way ANOVA with post-hoc Tukey test?  I think this needs clarifying to make it more evident to the reader.

Answer: We apologize for the mistake made. The analysis you use is one-way ANOVA.

- The discussion section contains many un-necessary results and there is need to avoid adding too many results in discussion section.

Answer: Following your recommendations, corrections have been made to the discussion in the revised manuscript.

Comments on the Quality of English Language

Please recheck the paper for typos.

Answer: We apologize for the typos made.

Corrections were also made to the text based on the comments and recommendations of the other two reviewers.

Sincerely yours,

Dr. Emilia Apostolova

Reviewer 2 Report

Comments and Suggestions for Authors

The study titled "Exploring Nitric Oxide as a Regulator in Salt Tolerance: Insights into Photosynthetic Efficiency in Maize" demonstrates the crucial role of nitric oxide (NO) in mitigating salt-induced stress on photosynthesis in maize (Zea mays L. Kerala). The research investigates the protective effects of varying concentrations of sodium nitroprusside (SNP), a NO donor, on the functionality of the photosynthetic apparatus under salt stress conditions.

The research suggests an optimal concentration range of NO, with approximately 50 - 63 nmoles NO/g FW in leaves, for achieving maximal protective effects against salt-induced stress. Overall, the study provides valuable insights into the role of NO as a regulator in salt tolerance and its potential application in improving photosynthetic efficiency in maize under adverse environmental conditions. The findings contribute to understanding of plant stress responses and offer promising informations for developing strategies to enhance crop resilience in saline environments.

The manuscript has the potential for publication in the journal Plants and needs the following adjustments and minor improvements:

3- lane 124: the results shown in Figure 1S do not correspond to the text in the results. In the graph, a significant decrease in the Chl/Car ratio is shown only in the 300SNP+Na Cl treatment.

6- lane 232: replace the word experiments with the word measurements

6-lane 235-237: move these sentences to material and methods. In the results, you should only write about the obtained results.

8. The parameter that represents dissipation per reaction center is unclear. "Reversed (RC/DI0)" is mentioned in the Figure 5 and in some parts of the text, but it should be changed to "DI0/RC".

8. „The additional information about the effects of SNP under salt stress on the PSII complex gives the dark relaxation of chlorophyll fluorescence by a signal after saturating light pulse in dark adapted leaves. Analysis of the curves provides details about the electron transfer from QA to plastoquinone [24]. Fluorescence signals in both treated and untreated maize plants can be fitted by two components, with the amplitude A1 (fast component) and A2 (slow component) with rate constant k1 and k2, respectively“- this should be written in the material and methods.

9 line 307: „Analyzing the flash-induced oxygen yields, we estimated the following kinetic parameters of oxygen evolution according to Kok’s model [61]: the active PSII centers in the initial reduced state (S0) (i.e., the initial S0–S1 state distribution of the PSII centers, S0% =100-S1 ) in the darkness, the misses and the double hits (ß).“- This should be written in the material and methods.

10. lane 333: It should be written treatments  instead of the word variety.

11. lane 364:  this sentence should be deleted because it is repeated “Salinity adversely affects plant growth and development, inhibiting various pro-cesses depending on plant species“

15. lane 604: „Differences among the various t were assessed“ what does „t“ mean? treatment?

In the discussion it is not necessary to write (Figure 1...Table 2......) when commenting on the obtained results.

Some parts of the material and methods should be better described... like... how many plants were used per treatment and how many leaves per plant? The extraction of pigments should also be better described.

The entire text should be read carefully and corrected...there are many mistakes like period, space...etc

The study titled "Exploring Nitric Oxide as a Regulator in Salt Tolerance: Insights into Photosynthetic Efficiency in Maize" demonstrates the crucial role of nitric oxide (NO) in mitigating salt-induced stress on photosynthesis in maize (Zea mays L. Kerala). The research investigates the protective effects of varying concentrations of sodium nitroprusside (SNP), a NO donor, on the functionality of the photosynthetic apparatus under salt stress conditions.

The research suggests an optimal concentration range of NO, with approximately 50 - 63 nmoles NO/g FW in leaves, for achieving maximal protective effects against salt-induced stress. Overall, the study provides valuable insights into the role of NO as a regulator in salt tolerance and its potential application in improving photosynthetic efficiency in maize under adverse environmental conditions. The findings contribute to understanding of plant stress responses and offer promising informations for developing strategies to enhance crop resilience in saline environments.

The manuscript has the potential for publication in the journal Plants and needs the following adjustments and minor improvements:

3- lane 124: the results shown in Figure 1S do not correspond to the text in the results. In the graph, a significant decrease in the Chl/Car ratio is shown only in the 300SNP+Na Cl treatment.

6- lane 232: replace the word experiments with the word measurements

6-lane 235-237: move these sentences to material and methods. In the results, you should only write about the obtained results.

8. The parameter that represents dissipation per reaction center is unclear. "Reversed (RC/DI0)" is mentioned in the Figure 5 and in some parts of the text, but it should be changed to "DI0/RC".

8. „The additional information about the effects of SNP under salt stress on the PSII complex gives the dark relaxation of chlorophyll fluorescence by a signal after saturating light pulse in dark adapted leaves. Analysis of the curves provides details about the electron transfer from QA to plastoquinone [24]. Fluorescence signals in both treated and untreated maize plants can be fitted by two components, with the amplitude A1 (fast component) and A2 (slow component) with rate constant k1 and k2, respectively“- this should be written in the material and methods.

9 line 307: „Analyzing the flash-induced oxygen yields, we estimated the following kinetic parameters of oxygen evolution according to Kok’s model [61]: the active PSII centers in the initial reduced state (S0) (i.e., the initial S0–S1 state distribution of the PSII centers, S0% =100-S1 ) in the darkness, the misses and the double hits (ß).“- This should be written in the material and methods.

10. lane 333: It should be written treatments  instead of the word variety.

11. lane 364:  this sentence should be deleted because it is repeated “Salinity adversely affects plant growth and development, inhibiting various pro-cesses depending on plant species“

15. lane 604: „Differences among the various t were assessed“ what does „t“ mean? treatment?

In the discussion it is not necessary to write (Figure 1...Table 2......) when commenting on the obtained results.

Some parts of the material and methods should be better described... like... how many plants were used per treatment and how many leaves per plant? The extraction of pigments should also be better described.

The entire text should be read carefully and corrected...there are many mistakes like period, space...etc

The study titled "Exploring Nitric Oxide as a Regulator in Salt Tolerance: Insights into Photosynthetic Efficiency in Maize" demonstrates the crucial role of nitric oxide (NO) in mitigating salt-induced stress on photosynthesis in maize (Zea mays L. Kerala). The research investigates the protective effects of varying concentrations of sodium nitroprusside (SNP), a NO donor, on the functionality of the photosynthetic apparatus under salt stress conditions.

The research suggests an optimal concentration range of NO, with approximately 50 - 63 nmoles NO/g FW in leaves, for achieving maximal protective effects against salt-induced stress. Overall, the study provides valuable insights into the role of NO as a regulator in salt tolerance and its potential application in improving photosynthetic efficiency in maize under adverse environmental conditions. The findings contribute to understanding of plant stress responses and offer promising informations for developing strategies to enhance crop resilience in saline environments.

The manuscript has the potential for publication in the journal Plants and needs the following adjustments and minor improvements:

3- lane 124: the results shown in Figure 1S do not correspond to the text in the results. In the graph, a significant decrease in the Chl/Car ratio is shown only in the 300SNP+Na Cl treatment.

6- lane 232: replace the word experiments with the word measurements

6-lane 235-237: move these sentences to material and methods. In the results, you should only write about the obtained results.

8. The parameter that represents dissipation per reaction center is unclear. "Reversed (RC/DI0)" is mentioned in the Figure 5 and in some parts of the text, but it should be changed to "DI0/RC".

8. „The additional information about the effects of SNP under salt stress on the PSII complex gives the dark relaxation of chlorophyll fluorescence by a signal after saturating light pulse in dark adapted leaves. Analysis of the curves provides details about the electron transfer from QA to plastoquinone [24]. Fluorescence signals in both treated and untreated maize plants can be fitted by two components, with the amplitude A1 (fast component) and A2 (slow component) with rate constant k1 and k2, respectively“- this should be written in the material and methods.

9 line 307: „Analyzing the flash-induced oxygen yields, we estimated the following kinetic parameters of oxygen evolution according to Kok’s model [61]: the active PSII centers in the initial reduced state (S0) (i.e., the initial S0–S1 state distribution of the PSII centers, S0% =100-S1 ) in the darkness, the misses and the double hits (ß).“- This should be written in the material and methods.

10. lane 333: It should be written treatments  instead of the word variety.

11. lane 364:  this sentence should be deleted because it is repeated “Salinity adversely affects plant growth and development, inhibiting various pro-cesses depending on plant species“

15. lane 604: „Differences among the various t were assessed“ what does „t“ mean? treatment?

In the discussion it is not necessary to write (Figure 1...Table 2......) when commenting on the obtained results.

Some parts of the material and methods should be better described... like... how many plants were used per treatment and how many leaves per plant? The extraction of pigments should also be better described.

The entire text should be read carefully and corrected...there are many mistakes like period, space...etc

The study titled "Exploring Nitric Oxide as a Regulator in Salt Tolerance: Insights into Photosynthetic Efficiency in Maize" demonstrates the crucial role of nitric oxide (NO) in mitigating salt-induced stress on photosynthesis in maize (Zea mays L. Kerala). The research investigates the protective effects of varying concentrations of sodium nitroprusside (SNP), a NO donor, on the functionality of the photosynthetic apparatus under salt stress conditions.

The research suggests an optimal concentration range of NO, with approximately 50 - 63 nmoles NO/g FW in leaves, for achieving maximal protective effects against salt-induced stress. Overall, the study provides valuable insights into the role of NO as a regulator in salt tolerance and its potential application in improving photosynthetic efficiency in maize under adverse environmental conditions. The findings contribute to understanding of plant stress responses and offer promising informations for developing strategies to enhance crop resilience in saline environments.

The manuscript has the potential for publication in the journal Plants and needs the following adjustments and minor improvements:

3- lane 124: the results shown in Figure 1S do not correspond to the text in the results. In the graph, a significant decrease in the Chl/Car ratio is shown only in the 300SNP+Na Cl treatment.

6- lane 232: replace the word experiments with the word measurements

6-lane 235-237: move these sentences to material and methods. In the results, you should only write about the obtained results.

8. The parameter that represents dissipation per reaction center is unclear. "Reversed (RC/DI0)" is mentioned in the Figure 5 and in some parts of the text, but it should be changed to "DI0/RC".

8. „The additional information about the effects of SNP under salt stress on the PSII complex gives the dark relaxation of chlorophyll fluorescence by a signal after saturating light pulse in dark adapted leaves. Analysis of the curves provides details about the electron transfer from QA to plastoquinone [24]. Fluorescence signals in both treated and untreated maize plants can be fitted by two components, with the amplitude A1 (fast component) and A2 (slow component) with rate constant k1 and k2, respectively“- this should be written in the material and methods.

9 line 307: „Analyzing the flash-induced oxygen yields, we estimated the following kinetic parameters of oxygen evolution according to Kok’s model [61]: the active PSII centers in the initial reduced state (S0) (i.e., the initial S0–S1 state distribution of the PSII centers, S0% =100-S1 ) in the darkness, the misses and the double hits (ß).“- This should be written in the material and methods.

10. lane 333: It should be written treatments  instead of the word variety.

11. lane 364:  this sentence should be deleted because it is repeated “Salinity adversely affects plant growth and development, inhibiting various pro-cesses depending on plant species“

15. lane 604: „Differences among the various t were assessed“ what does „t“ mean? treatment?

In the discussion it is not necessary to write (Figure 1...Table 2......) when commenting on the obtained results.

Some parts of the material and methods should be better described... like... how many plants were used per treatment and how many leaves per plant? The extraction of pigments should also be better described.

The entire text should be read carefully and corrected...there are many mistakes like period, space...etc

The study titled "Exploring Nitric Oxide as a Regulator in Salt Tolerance: Insights into Photosynthetic Efficiency in Maize" demonstrates the crucial role of nitric oxide (NO) in mitigating salt-induced stress on photosynthesis in maize (Zea mays L. Kerala). The research investigates the protective effects of varying concentrations of sodium nitroprusside (SNP), a NO donor, on the functionality of the photosynthetic apparatus under salt stress conditions.

The research suggests an optimal concentration range of NO, with approximately 50 - 63 nmoles NO/g FW in leaves, for achieving maximal protective effects against salt-induced stress. Overall, the study provides valuable insights into the role of NO as a regulator in salt tolerance and its potential application in improving photosynthetic efficiency in maize under adverse environmental conditions. The findings contribute to understanding of plant stress responses and offer promising informations for developing strategies to enhance crop resilience in saline environments.

The manuscript has the potential for publication in the journal Plants and needs the following adjustments and minor improvements:

3- lane 124: the results shown in Figure 1S do not correspond to the text in the results. In the graph, a significant decrease in the Chl/Car ratio is shown only in the 300SNP+Na Cl treatment.

6- lane 232: replace the word experiments with the word measurements

6-lane 235-237: move these sentences to material and methods. In the results, you should only write about the obtained results.

8. The parameter that represents dissipation per reaction center is unclear. "Reversed (RC/DI0)" is mentioned in the Figure 5 and in some parts of the text, but it should be changed to "DI0/RC".

8. „The additional information about the effects of SNP under salt stress on the PSII complex gives the dark relaxation of chlorophyll fluorescence by a signal after saturating light pulse in dark adapted leaves. Analysis of the curves provides details about the electron transfer from QA to plastoquinone [24]. Fluorescence signals in both treated and untreated maize plants can be fitted by two components, with the amplitude A1 (fast component) and A2 (slow component) with rate constant k1 and k2, respectively“- this should be written in the material and methods.

9 line 307: „Analyzing the flash-induced oxygen yields, we estimated the following kinetic parameters of oxygen evolution according to Kok’s model [61]: the active PSII centers in the initial reduced state (S0) (i.e., the initial S0–S1 state distribution of the PSII centers, S0% =100-S1 ) in the darkness, the misses and the double hits (ß).“- This should be written in the material and methods.

10. lane 333: It should be written treatments  instead of the word variety.

11. lane 364:  this sentence should be deleted because it is repeated “Salinity adversely affects plant growth and development, inhibiting various pro-cesses depending on plant species“

15. lane 604: „Differences among the various t were assessed“ what does „t“ mean? treatment?

In the discussion it is not necessary to write (Figure 1...Table 2......) when commenting on the obtained results.

Some parts of the material and methods should be better described... like... how many plants were used per treatment and how many leaves per plant? The extraction of pigments should also be better described.

The entire text should be read carefully and corrected...there are many mistakes like period, space...etc

The study titled "Exploring Nitric Oxide as a Regulator in Salt Tolerance: Insights into Photosynthetic Efficiency in Maize" demonstrates the crucial role of nitric oxide (NO) in mitigating salt-induced stress on photosynthesis in maize (Zea mays L. Kerala). The research investigates the protective effects of varying concentrations of sodium nitroprusside (SNP), a NO donor, on the functionality of the photosynthetic apparatus under salt stress conditions.

The research suggests an optimal concentration range of NO, with approximately 50 - 63 nmoles NO/g FW in leaves, for achieving maximal protective effects against salt-induced stress. Overall, the study provides valuable insights into the role of NO as a regulator in salt tolerance and its potential application in improving photosynthetic efficiency in maize under adverse environmental conditions. The findings contribute to understanding of plant stress responses and offer promising informations for developing strategies to enhance crop resilience in saline environments.

The manuscript has the potential for publication in the journal Plants and needs the following adjustments and minor improvements:

3- lane 124: the results shown in Figure 1S do not correspond to the text in the results. In the graph, a significant decrease in the Chl/Car ratio is shown only in the 300SNP+Na Cl treatment.

6- lane 232: replace the word experiments with the word measurements

6-lane 235-237: move these sentences to material and methods. In the results, you should only write about the obtained results.

8. The parameter that represents dissipation per reaction center is unclear. "Reversed (RC/DI0)" is mentioned in the Figure 5 and in some parts of the text, but it should be changed to "DI0/RC".

8. „The additional information about the effects of SNP under salt stress on the PSII complex gives the dark relaxation of chlorophyll fluorescence by a signal after saturating light pulse in dark adapted leaves. Analysis of the curves provides details about the electron transfer from QA to plastoquinone [24]. Fluorescence signals in both treated and untreated maize plants can be fitted by two components, with the amplitude A1 (fast component) and A2 (slow component) with rate constant k1 and k2, respectively“- this should be written in the material and methods.

9 line 307: „Analyzing the flash-induced oxygen yields, we estimated the following kinetic parameters of oxygen evolution according to Kok’s model [61]: the active PSII centers in the initial reduced state (S0) (i.e., the initial S0–S1 state distribution of the PSII centers, S0% =100-S1 ) in the darkness, the misses and the double hits (ß).“- This should be written in the material and methods.

10. lane 333: It should be written treatments  instead of the word variety.

11. lane 364:  this sentence should be deleted because it is repeated “Salinity adversely affects plant growth and development, inhibiting various pro-cesses depending on plant species“

15. lane 604: „Differences among the various t were assessed“ what does „t“ mean? treatment?

In the discussion it is not necessary to write (Figure 1...Table 2......) when commenting on the obtained results.

Some parts of the material and methods should be better described... like... how many plants were used per treatment and how many leaves per plant? The extraction of pigments should also be better described.

The entire text should be read carefully and corrected...there are many mistakes like period, space...etc

Author Response

Report to the comments of reviewer 2 on the manuscript (plants-2974020) titled  “ Exploring Nitric Oxide as a Regulator in Salt Tolerance: Insights into Photosynthetic Efficiency in Maize” by Rashkov et al.

The authors would like to thank the reviewer for constructive and insightful comments about this work. We considered all comments and suggestions to be justified, and corrected the manuscript accordingly. Please, find the detailed list of all edits below. The newly edited text parts are indicated with red letters.

Comments and Suggestions for Authors

The study titled "Exploring Nitric Oxide as a Regulator in Salt Tolerance: Insights into Photosynthetic Efficiency in Maize" demonstrates the crucial role of nitric oxide (NO) in mitigating salt-induced stress on photosynthesis in maize (Zea mays L. Kerala). The research investigates the protective effects of varying concentrations of sodium nitroprusside (SNP), a NO donor, on the functionality of the photosynthetic apparatus under salt stress conditions.

The research suggests an optimal concentration range of NO, with approximately 50 - 63 nmoles NO/g FW in leaves, for achieving maximal protective effects against salt-induced stress. Overall, the study provides valuable insights into the role of NO as a regulator in salt tolerance and its potential application in improving photosynthetic efficiency in maize under adverse environmental conditions. The findings contribute to understanding of plant stress responses and offer promising informations for developing strategies to enhance crop resilience in saline environments.

The manuscript has the potential for publication in the journal Plants and needs the following adjustments and minor improvements:

3- lane 124: the results shown in Figure 1S do not correspond to the text in the results. In the graph, a significant decrease in the Chl/Car ratio is shown only in the 300SNP+Na Cl treatment.

Answer: Figure 3S shows the changes in the Car/Chl ratio after treatment with NaCl and co-treatment with SNP and NaCl. A decrease in this ratio was observed when the plants were treated with NaCl alone and co-teatment NaCl and 300 µM SNP (there are statistically significant differences, the letters are b)

6- lane 232: replace the word experiments with the word measurements

Answer: The correction was made in the revised manuscript.

6-lane 235-237: move these sentences to material and methods. In the results, you should only write about the obtained results.

Answer: The sentences were moved in material and methods.

  1. The parameter that represents dissipation per reaction center is unclear. "Reversed (RC/DI0)" is mentioned in the Figure 5 and in some parts of the text, but it should be changed to "DI0/RC".

Answer: We use the reversed parameter RC/Dio, because its values are significantly greater than the values of the other studied JIP parameters, and plotting the values of the DIo/RC parameter on the Figure 5 will not clearly show the differences in the other studied JIP parameters. If you still think that it is better to use the DIo/RC parameter instead of RC/DIo we can change it both in the figure and in the text.

  1. „The additional information about the effects of SNP under salt stress on the PSII complex gives the dark relaxation of chlorophyll fluorescence by a signal after saturating light pulse in dark adapted leaves. Analysis of the curves provides details about the electron transfer from QA to plastoquinone [24]. Fluorescence signals in both treated and untreated maize plants can be fitted by two components, with the amplitude A1 (fast component) and A2 (slow component) with rate constant k1 and k2, respectively“- this should be written in the material and methods.

Answer: The sentences were moved in Material and Methods.

9 line 307: „Analyzing the flash-induced oxygen yields, we estimated the following kinetic parameters of oxygen evolution according to Kok’s model [61]: the active PSII centers in the initial reduced state (S0) (i.e., the initial S0–S1 state distribution of the PSII centers, S0% =100-S1 ) in the darkness, the misses and the double hits (ß).“- This should be written in the material and methods.

Answer: The sentences were moved in Material and methods.

  1. lane 333: It should be written treatments  instead of the word variety.

Answer: A correction was made in the revised manuscript.

  1. lane 364:  this sentence should be deleted because it is repeated “Salinity adversely affects plant growth and development, inhibiting various pro-cesses depending on plant species“

Answer: The sentence was deleted in the revised manuscript.

  1. lane 604: „Differences among the various t were assessed“ what does „t“ mean? treatment?

Answer: A correction was made in the revised manuscript.

In the discussion it is not necessary to write (Figure 1...Table 2......) when commenting on the obtained results.

Answer:  Referring to Figures of Tables in the discussion is a common practice and we believe that this makes the discussion easier for the readers to understand.

Some parts of the material and methods should be better described... like... how many plants were used per treatment and how many leaves per plant? The extraction of pigments should also be better described.

Answer: Corrections were made in the revised manuscript.

The entire text should be read carefully and corrected...there are many mistakes like period, space

Answer: We are sorry for the mistakes made. We have carefully reviewed the manuscript and hope there are no more mistakes.

Sincerely yours,

Dr. Emilia Apostolova

Reviewer 3 Report

Comments and Suggestions for Authors

14.04.2024

Review of the manuscript entitled: “Exploring nitric oxide as a regulator in salt tolerance: insights into photosynthetic efficiency in maize”

                Salinity is a crucial stressor which has been enhanced due to  environmental pollution and climate changes. The elevated salinity decreases crops’ yield affecting roots absorption and photosynthesis. The salt ions hamper cell membranes and cause an oxidative stress leading to decrease in photosynthetic capacity and growth. The present study provides evidence that nitric oxide applied into maize plants from sodium nitroprusside (SNP) reduces effectively the negative impact of NaCl on photosynthetic pigments concentration, Car/Chl ratio, cellular ultrastucture, enzymes and pigments in photosystems. Additionally, The SNP foliar application improves cell membrane permeability under salt stress. These positive effects of NO on the maize leaf structure, pigment content and photochemistry depend on SNP concentration. The most effective protection against oxidative stress induced by salinity provides the application of SNP at the moderate concentrations compared with the lowest and highest concentrations. In this study, numerous structural, physiological, and biophysical markers were employed to demonstrate that NO alleviates the stress effects on maize leaves induced by elevated salinity. The comprehensive analysis of these markers collectively supports the conclusion that, at moderate concentrations, SNP proves beneficial for maize by mitigating the adverse effects of elevated salinity on photosynthesis.

In “Methods”, the chemical process leading to emission of NO from SNP solution in greenhouse or in natural conditions should be elucidated.  There is no information how many plants were cultivated or how was the density of plantation (“Material and methods”). The equation to calculate RFd can be provided in “Room temperature chlorophyll fluorescence”. Many JIP parameters were used to compare the plants exposed to the different treatments with NaCl and/or SNP. The biological interpretation of these biophysical parameters has sometimes been difficult (lines 259-263 and 307-314). If it is possible, these results may be more clearly presented when their physiological significance in photosynthesis is elucidated.

Nitrogen oxide (NO) is a substrate in photochemical reactions within the troposphere where NO2 is synthetized. Under visible light, NO2 participates in photochemical reactions that produce ozone, a strong oxidant, known to damage the photosynthetic apparatus. Thus, it becomes important to address whether the widespread application of sodium nitroprusside into crops could increase O3 concentration and consequently reduce photosynthesis and crops productivity. This issue may be regarded as beyond the scope of this study, but on the other hand, it seems that it should be shortly discussed to assess the practical considerations regarding the application of NO.

This study has been well designed and the hypotheses and conclusions have been supported by results. The numerous markers provided insights into the role of NO in alleviation of negative impacts of enhanced salinity on the photosynthetic apparatus and processes. I recommend this manuscript to be published after a minor revision. 

Small concerns:

Line 37: “…stroma closure?”, it should be: “…stomata closure…”

In figure 1 the name of the statistical test should be given.

Figure 2: The name of the statistical test is lacking.

Line 202: “n” should be deleted.

Line 419: the parenthesis should be deleted.

Author Response

Report to the comments of reviewer 3 on the manuscript (plants-2974020) titled  “Exploring Nitric Oxide as a Regulator in Salt Tolerance: Insights into Photosynthetic Efficiency in Maize” by Rashkov et al.

The authors would like to thank the reviewer for constructive and insightful comments about this work. We considered all comments and suggestions to be justified, and corrected the manuscript accordingly. Please, find the detailed list of all edits below. The newly edited text parts are indicated with red letters.

Review of the manuscript entitled: “Exploring nitric oxide as a regulator in salt tolerance: insights into photosynthetic efficiency in maize

                Salinity is a crucial stressor which has been enhanced due to environmental pollution and climate changes. The elevated salinity decreases crops’ yield affecting roots absorption and photosynthesis. The salt ions hamper cell membranes and cause an oxidative stress leading to decrease in photosynthetic capacity and growth. The present study provides evidence that nitric oxide applied into maize plants from sodium nitroprusside (SNP) reduces effectively the negative impact of NaCl on photosynthetic pigments concentration, Car/Chl ratio, cellular ultrastucture, enzymes and pigments in photosystems. Additionally, The SNP foliar application improves cell membrane permeability under salt stress. These positive effects of NO on the maize leaf structure, pigment content and photochemistry depend on SNP concentration. The most effective protection against oxidative stress induced by salinity provides the application of SNP at the moderate concentrations compared with the lowest and highest concentrations. In this study, numerous structural, physiological, and biophysical markers were employed to demonstrate that NO alleviates the stress effects on maize leaves induced by elevated salinity. The comprehensive analysis of these markers collectively supports the conclusion that, at moderate concentrations, SNP proves beneficial for maize by mitigating the adverse effects of elevated salinity on photosynthesis.

In “Methods”, the chemical process leading to emission of NO from SNP solution in greenhouse or in natural conditions should be elucidated.  There is no information how many plants were cultivated or how was the density of plantation (“Material and methods”). The equation to calculate RFd can be provided in “Room temperature chlorophyll fluorescence”. Many JIP parameters were used to compare the plants exposed to the different treatments with NaCl and/or SNP. The biological interpretation of these biophysical parameters has sometimes been difficult (lines 259-263 and 307-314). If it is possible, these results may be more clearly presented when their physiological significance in photosynthesis is elucidated.

Answer: The authors are grateful for helpful suggestions and have made appropriate changes in the revised manuscript.

Nitrogen oxide (NO) is a substrate in photochemical reactions within the troposphere where NO2 is synthetized. Under visible light, NO2 participates in photochemical reactions that produce ozone, a strong oxidant, known to damage the photosynthetic apparatus. Thus, it becomes important to address whether the widespread application of sodium nitroprusside into crops could increase O3 concentration and consequently reduce photosynthesis and crops productivity. This issue may be regarded as beyond the scope of this study, but on the other hand, it seems that it should be shortly discussed to assess the practical considerations regarding the application of NO.

This study has been well designed and the hypotheses and conclusions have been supported by results. The numerous markers provided insights into the role of NO in alleviation of negative impacts of enhanced salinity on the photosynthetic apparatus and processes. I recommend this manuscript to be published after a minor revision. 

Small concerns:

Line 37: “…stroma closure?”, it should be: “…stomata closure…”

In figure 1 the name of the statistical test should be given.

Figure 2: The name of the statistical test is lacking.

Line 202: “n” should be deleted.

Line 419: the parenthesis should be deleted.

Answer: All these remarks are corrected in the revised manuscript.

Sincerely yours,

Dr. Emilia Apostolova
